# DCT-CryptoNets: Scaling Private Inference in the Frequency Domain

**Arjun Roy**
Purdue University
roy208@purdue.edu

**Kaushik Roy**
Purdue University
kaushik@purdue.edu

## Abstract

The convergence of fully homomorphic encryption (FHE) and machine learning offers unprecedented opportunities for private inference of sensitive data. FHE enables computation directly on encrypted data, safeguarding the entire machine learning pipeline, including data and model confidentiality. However, existing FHE-based implementations for deep neural networks face significant challenges in computational cost, latency, and scalability, limiting their practical deployment. This paper introduces DCT-CryptoNets, a novel approach that operates directly in the frequency-domain to reduce the burden of computationally expensive non-linear activations and homomorphic bootstrap operations during private inference. It does so by utilizing the discrete cosine transform (DCT), commonly employed in JPEG encoding, which has inherent compatibility with remote computing services where images are generally stored and transmitted in this encoded format. DCT-CryptoNets demonstrates a substantial latency reductions of up to $5.3\times$ compared to prior work on benchmark image classification tasks. Notably, it demonstrates inference on the ImageNet dataset within 2.5 hours (down from 12.5 hours on equivalent 96-thread compute resources). Furthermore, by *learning* perceptually salient low-frequency information DCT-CryptoNets improves the reliability of encrypted predictions compared to RGB-based networks by reducing error accumulating homomorphic bootstrap operations. DCT-CryptoNets also demonstrates superior scalability to RGB-based networks by further reducing computational cost as image size increases. This study demonstrates a promising avenue for achieving efficient and practical private inference of deep learning models on high resolution images seen in real-world applications.[*]

## 1 Introduction

Escalating privacy and security concerns in machine learning have fueled exploration of private inference techniques. These approaches aim to protect both sensitive data and model confidentiality while maintaining a high quality user experience, especially when related to inference latency and accuracy. To address these challenges, private inference solutions based on cryptographic principles, such as fully homomorphic encryption (FHE) have been proposed. However, FHE's strong security guarantee often comes with significant computational overhead and latency, creating barriers to widespread adoption.

Early work on fully homomorphic encrypted neural networks (FHENNs) faced limitations in both latency and accuracy. Due to the limited native operations of homomorphic encryption, primarily addition and multiplication, many prior methods resorted to approximating non-linear activation functions using polynomials (Dowlin et al., 2016; Brutzkus et al., 2018; Chou et al., 2018). However, this approach introduces accuracy degradation as networks deepen, due to the cumulative effect of approximation errors. Newer homomorphic encryption schemes like TFHE (FHE over the Torus) (Chillotti et al., 2019) can handle non-linear activation functions without the need for approximations. Still, optimizing the efficiency of homomorphic operations (HOPs), which include convolutions and non-linear activations in the context of neural networks, remains a key area of research.

---

[*]Code is available at https://github.com/ar-roy/dct-cryptonets

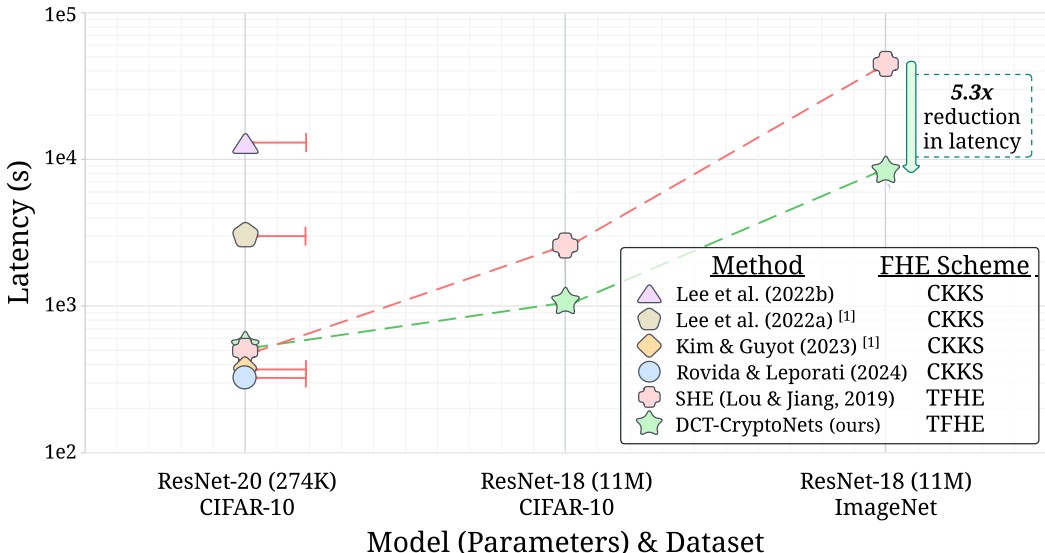

Figure 1: Scalability of state of the art image classification methods in FHENN. DCT-CryptoNets is able to reduce latency by up to 5.3× compared to SHE (Lou & Jiang, 2019), the only other published method able to infer on ImageNet. To ensure a fair comparison, SHE latency values were normalized to the same computational resources as DCT-CryptoNets (96-threads). CKKS-based methods have difficulty scaling to larger networks and datasets due to their highly approximate nature.[†]

To reduce the computational burden of HOPs in FHENNs, previous work has investigated strategies for optimizing convolutions in the encrypted domain (Lou et al., 2020; Lee et al., 2022b; Kim & Guyot, 2023; Ran et al., 2023; Rovida & Leporati, 2024) and designing architectures that minimize the use of non-linear activations (Ghodsi et al., 2020; Jha et al., 2021). In this work, we prioritize optimizing non-linear activations by operating in the frequency domain. This approach is motivated by the significant computational cost of non-linear activations in FHE (e.g., ReLU operations consuming ~32.6% of total inference time (Lee et al., 2022b)), and the fact that non-linear activations scale primarily with the spatial extent of an input. As demonstrated in Jha et al. (2021), non-linear activations are disproportionately concentrated in the early layers of networks where spatial dimensionality is the largest (e.g., 48% within the first quarter of a ResNet-18 network).

While notable advancements have been made, many FHENN methods remain tailored to small networks and images (e.g., 32×32). Scaling FHENN to ImageNet and larger networks presents a formidable challenge. Increasing both network size and image resolution introduces latency from not only convolutions but also non-linear activations, which are "free" in unencrypted networks, and an expensive homomorphic bootstrapping operation (Gentry, 2010). Homomorphic bootstrap is used to "reset" noise in the ciphertext that accumulate for every encrypted operation (addition and multiplication), which can eventually corrupt the ciphertext making decryption impossible. Approximately 31.6% of total inference time is spent on such bootstrapping operations (Lee et al., 2022b). While attempts have been made to implement FHENN on large-scale datasets such as ImageNet, these efforts highlight the substantial trade-offs involved: either prohibitive inference latency (e.g., 2.5 days for ResNet-18 in Lou & Jiang (2019)) or severely limited encryption scope (e.g., encrypting only 8 layers in ResNet-18 in Kim & Guyot (2023)) as shown in Figure 1.

This work introduces DCT-CryptoNets, a novel framework that addresses the computational challenges of fully homomorphic encrypted neural networks (FHENNs). Traditional convolutional neural networks operate on raw pixel data, learning features from *spatial intensity variations*. Instead, we utilize Discrete Cosine Transforms (DCT) to represent images in the frequency domain, enabling our models to learn features from the *rate of change in intensities* (Gueguen et al., 2018; Ehrlich & Davis, 2019). This not only aligns with the human visual system's differential sensitivity to perceptually

---

[†]Lee et al. (2022a) scale to ResNet-110 (1.7M parameters). Kim & Guyot (2023) scale to a Plain-18 network (ResNet-18 without skip connections) but only encrypt the last 8 layers when running on ImageNet.

relevant low-frequency information (Luo et al., 2022), but the reduction in spatial dimensionality by operating in the frequency domain also mitigates the computational burden of non-linear activations. DCT-CryptoNets yields two additional advantages: (1) it reduces computationally expensive homomorphic bootstrapping, as fewer non-linear activations result in less ciphertext noise accumulation, and (2) it improves the reliability of encrypted accuracy through the reduction of bootstrap operations which accumulate approximation errors. Additionally, the emphasis on low-frequency information has the potential to boost accuracy compared to traditional RGB-based CNNs due to focusing on visually salient information. Overall, our method demonstrates substantial latency improvements in convolution-based FHENNs across a range of image classification datasets, culminating in a novel demonstration of ImageNet inference within 2.5 hours – a significant advancement over SHE (Lou & Jiang, 2019) that required 12.5 hours on equivalent compute resources (previously reported as 2.5 days). These results underscore the potential of DCT-CryptoNets for accelerating privacy-preserving deep learning applications.

Our approach makes the following contributions:

- We propose DCT-CryptoNets to achieve significant latency improvements in image classification FHENNs by utilizing DCT. Our focus on low-frequency components reduces the impact of computationally expensive non-linear activations and homomorphic bootstrapping, resulting in a $5.3\times$ latency reduction on ImageNet, while maintaining or even boosting accuracy compared to RGB-based networks due to the focus on perceptually salient information.

- We show that DCT-CryptoNets also bolsters the reliability of encrypted accuracy. By curtailing the need for homomorphic bootstrap operations, these optimizations curb error accumulation. This results in a notable reduction in encrypted accuracy variability (e.g., from ±2.5% to ±1.0% on ImageNet).

- We show that DCT-CryptoNets exhibits superior scalability with increasing image resolution. This is evident in the amplified reduction of homomorphic operations (HOPs) observed for larger images (e.g., a 33.1% vs. 44.1% reduction in HOPs when applying DCT-based frequency optimizations to a $224\times224$ vs. $448\times448$ image respectively).

## 2 BACKGROUND AND RELATED WORK

### 2.1 CRYTOGRAPHIC PROTOCOLS FOR PRIVATE INFERENCE

Homomorphic encryption relies on lattice-based cryptography (Regev, 2005) and the Learning with Errors (LWE) problem (Lyubashevsky et al., 2010), leveraging the computational hardness of solving equations on high-dimensional lattices with added noise. Lattice-based cryptography has gained significant traction, with 3 out of the 4 finalists in the National Institute of Standards and Technology (NIST) post-quantum cryptography standardization being based on this approach.

Many state-of-the-art private inference techniques leverage hybrid, "interactive" approaches that combine homomorphic encryption with other cryptographic protocols such as multi-party computation (MPC) (Juvekar et al., 2018; Mishra et al., 2020; Knott et al., 2021). In MPC, multiple parties jointly compute a function on their private inputs without disclosing those inputs. While ensuring no party learns more than the final result, MPC still can leak information as communication between parties is necessary. Although robust, these methods still pose a security risk in low-trust environments. In contrast, "non-interactive" FHE-only methods offer superior data confidentiality. FHE ensures that neither raw data nor intermediate values are exposed in plaintext. This makes FHE particularly well-suited for scenarios with highly sensitive data and model parameters, especially in situations where there is severely limited trust between parties (see Appendix A.1). However, this enhanced privacy comes at the cost of increased computational overhead compared to MPC and hybrid approaches.

### 2.2 LIMITATIONS OF EXISTING FHENN SCHEMES

FHE schemes serve as the fundamental cryptographic building blocks for various application-specific methodologies. The choice of an FHE scheme involves a careful evaluation of the trade-offs inherent to each scheme to ensure optimal performance. FHENNs based on earlier schemes such as BFV (Fan & Vercauteren, 2012) and BGV (Brakerski et al., 2012) were known for their efficiency for encrypted

addition and multiplication. However, these methods often faced scalability challenges due to their reliance on polynomial approximations of activations (PAAs) and pooling operations. This reliance on PAAs introduces accuracy degradation due to the cumulative effect of approximation errors in deeper networks. To mitigate accuracy degradation, techniques to increase the polynomial degree of PAAs have been introduced (Hesamifard et al., 2019). Despite this, the computational cost of PAA-based non-linear activations increases exponentially with network depth, becoming a major bottleneck to latency in deeper networks (Lou & Jiang, 2019). A more recent scheme, CKKS (Cheon et al., 2017), offers a compelling advantage for neural network applications by operating directly on floating-point numbers and supporting single instruction multiple data (SIMD) operations. Nevertheless, CKKS inherits the same challenges associated with PAAs and also introduces additional accuracy concerns due to its reliance on approximate arithmetic, which can lead to significant error accumulation in deep networks.

Further efforts to optimize PAA-based methods have shown promise but remain limited. Lee et al. (2022b) introduced a binary-tree implementation for ReLU in the CKKS scheme, yet scalability is confined to smaller models like ResNet-20. Falcon (Lou et al., 2020) introduced fast homomorphic discrete Fourier transform (HDFT) for convolutions and fully connected layers in BFV, significantly reducing operations. However, this approach still suffers from limited accuracy (76.5% on CIFAR-10) and requires specialized convolution blocks, hindering broader applicability with existing CNN architectures. Kim & Guyot (2023) extended HDFT to CKKS, but only the final 8 layers of a Plain-18 network are encrypted when operating on ImageNet. Lee et al. (2022a) proposed removing the imaginary components of PAAs, thus enabling scalability up to ResNet-110 (1.7M parameters). While PAA-based methods have shown promise for smaller networks, their inherent constraints in handling increased depth and computational complexity hinder their scalability to deeper networks.

## 2.3 ADVANTAGES OF TFHE

In contrast to other HE schemes, TFHE (FHE over the Torus) (Chillotti et al., 2019) operates within the domain of the torus modulo 1, representing ciphertexts as elements of this structure. TFHE excels in performing fast and exact binary operations on encrypted bits, utilizing integer and logic gates. The foundation for these operations lies in the M-th cyclotomic polynomial, denoted as $\Phi(X)$. Its degree, represented by $N$, is typically chosen as a power of 2 for efficiency reasons, and simplifies $\Phi(X)$ to $X^N + 1$. By defining polynomial rings over real coefficients $R_N[X] := \mathbb{R}[X]/(X^N + 1)$ and integer coefficients $Z_N[X] := \mathbb{Z}[X]/(X^N + 1)$, we establish the Torus polynomial ring $\mathbb{T}_N[X] := R_N[X]/Z_N[X] = \mathbb{T}[X]/(X^N + 1)$ which forms a $\mathbb{Z}_N[X]$-module. This algebraic structure is fundamental to TFHE, facilitating both addition and external multiplication by polynomials of $\mathbb{Z}_N[X]$.

Unlike HE schemes that rely on PAAs, TFHE's ability to directly implement non-linear activations using Boolean/integer arithmetic is a key factor in scalability. Directly implementing non-linear activations avoids the accuracy degradation often associated with PAAs in deeper networks. SHE (Lou & Jiang, 2019), a TFHE-based FHENN, shows scalability to ImageNet. They do so by employing a bit-series representation and techniques like logarithmic weight quantization and bit-shift-based convolutions. However, SHE still suffers from prohibitive latency on ImageNet (e.g., ResNet-18 in 2.5 days, ShuffleNet in 5 hours per image). Furthermore, SHE utilizes a leveled TFHE scheme which necessitates higher multiplicative depths to avoid decryption failures as a network deepens. This higher multiplicative depth budget results in larger ciphertexts which consequently increases latency in these deeper networks.

In our approach, we apply programmable bootstrapping (PBS) mechanisms (Chillotti et al., 2021) to support arbitrarily deep neural networks and mitigate the limitations of leveled schemes. PBS serves two crucial purposes: (1) better ciphertext noise reduction by reordering rotation and key-switch ciphertext operations and (2) enabling homomorphic evaluation of any function expressible as a lookup-table. PBS's ability to homomorphically evaluate functions expressed as lookup-tables makes it well-suited for implementing non-linear activations.

## 2.4 HYPER-QUANTIZATION BACKGROUND

Hyper-quantization is a technique that improves the efficiency of neural networks by reducing the precision of numerical representations (weights and activations). This not only lowers memory

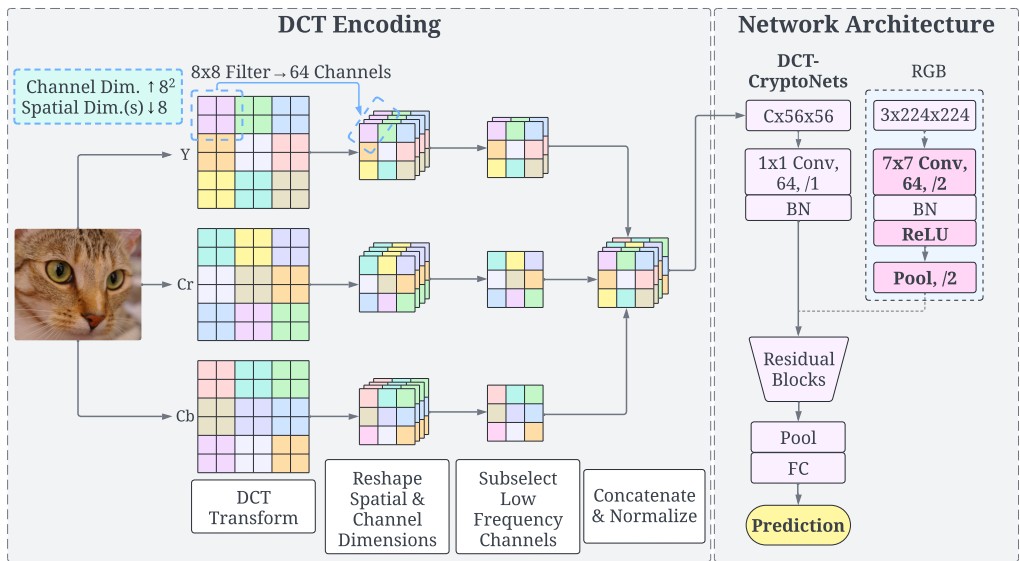

Figure 2: DCT-CryptoNets' frequency encoding (based on Xu et al. (2020)) and ResNet-18 network architecture. Modifications from an RGB-based network to DCT-CryptoNets are emphasized in bold and darker purple. These include kernel size and downsampling of the first convolution layer, as well as exclusion of both the ReLU operator and pooling layer after the first convolutional layer. This approach requires minimal modification of existing networks to utilize DCT, making conversion for many potential applications simple.

requirements and accelerates computation but also naturally aligns with the integer-based polynomial representation in TFHE, facilitating efficient computation within this encrypted domain. However, hyper-quantization often leads to a trade-off in accuracy as the limited representation can introduce errors.

Quantization-aware training (QAT) can help mitigate these accuracy trade-offs by simulating the effects of quantization during the training process. By constraining weights and activations to fixed-point representations, QAT intentionally introduces quantization noise during training, enabling the network to learn and adapt to these errors through gradient-based optimization (Nagel et al., 2021). In the context of uniform quantization, we convert a real value $r \in [\alpha, \beta]$ into a $b$-bit integer $q = \lfloor \frac{r}{S} + Z \rfloor$, where $S = \frac{\beta - \alpha}{2^b - 1}$ represents the quantization scale and Z is the zero-point. Previous efforts to quantize FHENNs have produced promising results. Notably, Stoian et al. (2023) achieved inference on a VGG-9 network using an 8-bit quantized TFHE scheme with programmable bootstrapping (PBS). Their approach achieved 87.5% accuracy on a VGG-9 network but still had significant latency of 18,000 seconds for inference on the CIFAR-10 dataset. More aggressive quantization can also yield latency reductions, such as a 40% decrease in runtime when moving from an 8-bit to a 2-bit BFV-based FHENN for CIFAR-10 (Legiest et al., 2023). Quantization-aware training facilitates the seamless conversion of our models into a TFHE-compatible framework, enabling us to maintain high accuracy while significantly reducing computational overhead.

## 2.5   FREQUENCY DOMAIN BACKGROUND

In a departure from prior private inference work and frequency-domain techniques (Bian et al., 2020; Lou et al., 2020; Kim & Guyot, 2023), our method learns features from the frequency-domain representation of images, enabling the network to learn from the rate of change of intensities. This is done through the utilization of the Discrete Cosine Transform (DCT) - a key component of JPEG compression, widely used for image transmission and storage (Gueguen et al., 2018; Ehrlich & Davis, 2019). This allows our method to be directly compatible with machine learning as a service (MLaaS) systems and other applications where image data is transmitted in compressed formats for remote analysis (see Appendix A.2). The 1-D DCT transform uses the following equation where $x_n$ represents a signal and $X_k$ represent the DCT coefficients transformed via sinusoidal bases. DCT in

Table 1: Comparison of multiply-accumulate (MAC), ReLU, programmable bootstrap (PBS) and total homomorphic operations (HOPs) for RGB-based and DCT-based ResNet-18 networks. DCT consistently reduces ReLUs, PBS and HOPs, even with varying numbers of retained low-frequency channels, enabling accuracy-driven selection. Notably, DCT's HOP reduction improves with higher image resolutions (e.g., 11% greater HOP reduction for 448×448 vs. 224×224 images) due to additional 13.9% reduction in PBS operations, showcasing its scalability.

| | | ImageNet-sized Images | | | | Large(er) Images | | | |
|---|---|---|---|---|---|---|---|---|---|
| | Dimension | #MACs | #ReLUs | #PBS | #HOPs | Dimension | #MACs | #ReLUs | #PBS | #HOPs |
| RGB | $3{\times}224^2$ | 1.82G | 2.31M | 22.4M | 1.85G | $3{\times}448^2$ | 7.29G | 9.23M | 89.3M | 7.67G |
| DCT | $6{\times}56^2$ | 1.70G | 1.51M | 15.5M | 1.24G | $6{\times}112^2$ | 6.82G | 6.02M | 50.7M | 4.29G |
| | $24{\times}56^2$ | 1.71G | 1.51M | 15.5M | 1.26G | $24{\times}112^2$ | 6.83G | 6.02M | 50.9M | 4.33G |
| | $48{\times}56^2$ | 1.71G | 1.51M | 15.6M | 1.26G | $48{\times}112^2$ | 6.85G | 6.02M | 51.2M | 4.35G |
| | $64{\times}56^2$ | 1.72G | 1.51M | 15.6M | 1.27G | $64{\times}112^2$ | 6.86G | 6.02M | 51.3M | 4.38G |
| | $192{\times}56^2$ | 1.74G | 1.51M | 15.7M | 1.28G | $192{\times}112^2$ | 6.97G | 6.02M | 51.7M | 4.41G |
| **Max Δ RGB → DCT** | | -6.54% | -34.8% | -30.0% | -33.1% | | -6.54% | -34.8% | -43.9% | -44.1% |

2-D is the same 1-D equation applied to the width and height dimensions.

$$X_k = \sum_{n=0}^{N-1} x_n \cos\left[\frac{k\pi}{N}\left(n+\frac{1}{2}\right)\right] \qquad k = 0, \ldots, N-1. \tag{1}$$

While DCT itself is lossless and reversible, by selectively retaining only low-frequency components we can reduce data requirements while preserving image classification accuracy (Xu et al., 2020). This approach leverages the human visual system's differential sensitivity to various frequency components, allowing models to focus on the most perceptible elements. Adversarially robust object recognition models designed to resist perceptible perturbations have been shown to primarily rely on this low-frequency information (Subramanian et al., 2023; Luo et al., 2022). Additionally, utilizing DCT components directly can improve the privacy preservation of recognition systems against various white-box and black-box attacks (Wang et al., 2022). While previous work, such as Mertens et al. (2024), has explored DCT for homomorphic image compression and decompression, our method uniquely retains the encrypted data in the frequency domain throughout the CNN inference process. The shift from high-spatial RGB to low-frequency DCT components in our training approach offers a dual advantage. By prioritizing perceptually relevant information, it holds the potential for improved image classification. Simultaneously, the spatial dimensionality reduction of DCT significantly lowers the computational demands of homomorphic inference, particularly by reducing the burden of non-linear activations and homomorphic bootstrap operations due to the fact that non-linear activations are disproportionately concentrated in network layers with high spatial dimensionality.

## 3 METHODOLOGY

DCT-CryptoNets is composed of two key pieces: (1) We propose network architectures for low-frequency DCT components coupled with strategic reductions in ReLU activations for more optimal computation in the encrypted domain. These architectural designs deliver efficiency gains compared to traditional RGB-based networks across various image resolutions, even in the challenging case of low-resolution inputs where direct DCT application may be less effective. (2) We present a framework for quantization-aware training on these low-frequency DCT components, enabling seamless encryption into a TFHE-compatible scheme. We present multiple ablations into the effect of quantization and cryptographic hyperparameters on accuracy and latency in the encrypted domain.

### 3.1 NETWORK ARCHITECTURE IN THE DCT DOMAIN WITH RELU REDUCTIONS

For encoding DCT frequency tensors from images, we first transform RGB images to the YCrCb color space, then apply 2D-DCT operations to generate frequency domain representations (see Figure 2). We adopt standard JPEG compression 8×8 filtering scheme, yielding 64 frequency components per

Table 2: Evaluating performance on low resolution images with a traditional ResNet-20 and proposed DCT-CryptoNets ResNet-20 on CIFAR-10. DCT-CryptoNets ResNet-20, with 24 retained low-frequency channels and 16×16 spatial input, achieves a 4% accuracy improvement over the baseline while maintaining latency.

| | Input Dimension | Model | Accuracy | Latency (s) |
|---|---|---|---|---|
| RGB | $3 \times 32^2$ | Traditional ResNet-20 | 86.5% | 603 |
| | $3 \times 32^2$ | DCT-CryptoNets ResNet-20 | 91.6% | 1,339 |
| DCT | $24 \times 8^2$ | DCT-CryptoNets ResNet-20 | 85.2% | 137 |
| | $48 \times 8^2$ | DCT-CryptoNets ResNet-20 | 86.5% | 144 |
| | $24 \times 16^2$ | DCT-CryptoNets ResNet-20 | 90.5% | 565 |
| | $48 \times 16^2$ | DCT-CryptoNets ResNet-20 | 90.4% | 567 |

YCrCb color space. This process effectively downsamples spatial dimensions (H, W) by a factor of 8 while increasing channel dimensions (C) by a factor of $8^2$. Subsequently, we sub-select a fixed portion of the frequency components, emphasizing luma (Y) components due to their higher perceptual relevance compared to chroma (Cr, Cb). This sub-selection of frequency components can be viewed as varying the level of "lossy-ness" of an image from a highly lossy 6-component to a loss-less 192-component representation. We demonstrate the impact of varying the level of sub-selected frequency components (6, 24, 48, 64, and 192) on the number of homomorphic operations (HOPs) and model accuracy. Notably, the total number of HOPs remains consistent regardless of the sub-selection of frequency components, allowing for flexibility in choosing an input representation that provides high accuracy (see Table 1). The DCT's filtering of spatial information into frequency components proves particularly advantageous as image size increases. We observe that DCT becomes increasingly valuable for HOP optimization as we scale to ImageNet-sized images and beyond. For 448×448 images, we observe a 44.1% reduction in HOPs compared to a 33.1% reduction for 224×224 images (see Table 1). While the percentage of ReLU operations removed remains constant due to the inherent structure of DCT components, this amplified HOP reduction at larger image sizes can be attributed to the decreased need for bootstrapping operations, a direct consequence of fewer ReLU activations. This highlights the scalability of DCT-CryptoNets in handling larger inputs.

Given the distinct dimensionality of frequency-domain features (reduced H and W, increased C), we modify the initial stride-2 convolution layer, commonly found in CNNs, with a stride-1 layer. The subsequent layer's channel size is then adjusted to accommodate the transformed input (see Figure 2). Further optimization involves pruning the ReLU activations following the channel-widening convolution layer. This reduction in ReLU operations does not significantly impact accuracy, as earlier-layer ReLUs typically exhibit lower importance than those in later layers (Jha et al., 2021). While the skip connection pruning introduced in Ghodsi et al. (2020) also reduces ReLU count, it incurs a much greater accuracy cost and also increase HOPs in TFHE-based networks (see Appendix A.3). Therefore, we opted for a network architecture that specifically incorporates first-layer ReLU pruning. Figure 2 shows both the process for encoding DCT frequency tensors as well as the network architecture needed for handling DCT tensors.

While DCT offers significant benefits for larger images, its direct application to lower resolutions can lead to excessive spatial dimension reduction. To address this challenge, we introduce a second specialized architecture, tailored for smaller inputs (e.g., 32×32 pixels). This adaptation utilizes a 4×4 DCT filter, resulting in 48 total frequency channels, and incorporates architectural adjustments to maintain spatial resolution and prevent information loss (see Appendix A.4 for an architecture deep-dive). Despite an increased parameter count due to expansion of the channel dimensions, DCT-CryptoNets achieves similar latency coupled with a 4% accuracy improvement over a traditional ResNet-20 on CIFAR-10 (DCT-CryptoNets $24 \times 16^2$ vs. Traditional $3 \times 32^2$ in Table 2). Furthermore, even with further spatial reduction from 16×16 to 8×8, we observe similar accuracy to a traditional ResNet-20 coupled with a 4.2× latency improvement. This demonstrates the effectiveness of DCT-CryptoNets, not only for high resolution images but for low resolution images as well.

## 3.2 TRAINING AND ENCRYPTING INTO A TFHE-COMPATIBLE SCHEME

DCT-CryptoNets employs a quantization-aware training (QAT) framework to learn low-frequency components. To achieve an optimal balance between accuracy and latency for our FHENNs, we systematically explored various quantization levels (see Appendix A.5). Our approach employs a 4-bit symmetric quantization scheme with a fixed zero-point for all tasks except ImageNet, where 5-bit quantization is used. These choices strike a balance between efficiency and accuracy, allowing us to perform the entire neural network computation directly on encrypted data within the TFHE framework.

To fully encrypt the trained neural network within the TFHE scheme, we employ an optimization process (Bergerat et al., 2023). This process balances three key cryptographic hyperparameters: (1) the circuit bit-width, defined as the minimum number of bits required to represent the largest integer arising during computation. This generally mimics the 4-bit or 5-bit integer used during QAT; (2) precision rounding for model accumulators, which implements a scaling factor used to remove least significant bits; and (3) the desired error probability for programmable bootstrapping (PBS) operations. Unlike polynomial approximation (PAAs) that introduce inherent approximation errors, PBS error probability is a tunable hyperparameter. It controls the likelihood of a small error value added to the look-up table (LUT) operation. By carefully tuning these hyperparameters, the optimization process determines the cryptosystem parameters that ensure efficient execution.

$$LUT[x] = \begin{cases} T[x], & \text{with probability } (1 - p_{err}) \\ T[x+k], & \text{with probability } p_{err}, \ k = (\pm 1, \pm 2, ...) \end{cases} \tag{2}$$

Determining optimal cryptographic hyperparameters involves empirical exploration due to the indirect control over the underlying cryptosystem parameters. Decreasing the PBS error probability or increasing the precision rounding threshold to require more bits necessitates additional homomorphic operations (HOPs). This presents a trade-off: while enhancing accuracy, it also increases latency. We notice that for simpler tasks, a precision rounding of 6 and a PBS error probability of 0.01 achieve a good balance. However, more complex tasks like ImageNet require further hyperparameter tuning to maintain accuracy. For instance, increasing precision rounding to 7 reduces bit truncation but leads to a higher number of HOPs, thus increasing latency (see Appendix A.6 for a detailed analysis).

## 4 RESULTS

### 4.1 EXPERIMENTAL SETUP

DCT-CryptoNets utilizes several tools and frameworks. Image conversion to the frequency domain is performed using libjpeg-turbo, Brevitas (Pappalardo, 2023) facilitates quantization-aware training, and Concrete-ML (Zama, 2022) enables encryption within the TFHE scheme. Neural network training is conducted on an RTX A40 GPU and FHENN latency measurements are obtained from dual AMD Ryzen Threadripper PRO 5965WX processors (96-threads total). Each network is trained on CIFAR-10, mini-ImageNet, Imagenette, and ImageNet, with varying configurations of DCT components. To assess FHENN accuracy, we leverage Concrete-ML's simulation accuracy feature, allowing us to efficiently estimate expected accuracy and gather valuable metrics on the compiled FHENN. Latency measurements are conducted by processing a single image at a time on a dedicated machine with no background tasks running, ensuring maximum utilization of all available threads. All training, quantization and cryptographic hyperparameters are also disclosed in Appendix A.7.

### 4.2 ACCURACY AND RELIABILITY ENHANCEMENTS WITH LOW-FREQUENCY INFORMATION

To demonstrate the impact of focusing on visually salient low-frequency information we evaluated model accuracy across CIFAR-10, mini-ImageNet, Imagenette, and ImageNet on upsampled $3 \times 224^2$ RGB inputs and their DCT-based representations with varying frequency components (see Table 3). Due to the computational cost of obtaining simulation accuracy results in Concrete-ML (approximately 8 seconds per ImageNet image), comprehensive validation on the full ImageNet validation set was impractical. To estimate the impact of encryption on accuracy, we employed statistical bootstrapping to generate 95% confidence intervals (CIs) for each model. This involved repeatedly (10K times) analyzing random combinations of subsets (20 subsets each with 200 random images) to

Table 3: Top-1 accuracy of datasets with varying levels of retained low-frequency components, ranging from a highly lossy 6-channel representation to a lossless 192-channel representation. Accuracies are reported as unencrypted plus or minus a 95% confidence interval generated by the encryption process as explained in Section 4.2. As DCT models per dataset share similar number of HOPs and cryptographic hyperparameters, their confidence intervals are consistent.

| | Model | Input Dimension | CIFAR-10 (10 classes) | mini-ImageNet (100 classes) | Imagenette (10 classes) | ImageNet (1K classes) |
|---|---|---|---|---|---|---|
| RGB | ResNet-18 | $3\times224^2$ | 92.4±2.4% | 88.9±2.4% | 88.5±2.0% | 66.1±2.5% |
| DCT | ResNet-18 | $6\times56^2$ | 89.7±0.6% | 81.6±1.8% | 84.3±0.4% | 55.5±1.0% |
| | ResNet-18 | $24\times56^2$ | 91.2±0.6% | 90.0±1.8% | **87.5±0.4%** | 65.8±1.0% |
| | ResNet-18 | $48\times56^2$ | 91.7±0.6% | **90.1±1.8%** | 86.4±0.4% | 66.1±1.0% |
| | ResNet-18 | $64\times56^2$ | **92.4±0.6%** | 89.5±1.8% | 86.5±0.4% | **66.3±1.0%** |
| | ResNet-18 | $192\times56^2$ | 92.3±0.6% | 89.7±1.8% | 83.8±0.4% | 64.1±1.0% |

determine accuracy variability. The resulting distribution of accuracies allowed us to estimate the 95% CI for each model, reflecting the expected variability due to encryption.

DCT-based networks consistently matched or outperformed the accuracy of their RGB counterparts, particularly on larger datasets (mini-ImageNet & ImageNet), due to their focus on perceptually relevant low-frequency information. Generally, retaining the top 64 low-frequency channels yielded the best accuracy. Suggesting that loss-less representations (192 channels) contain unnecessary high-frequency components that hinder learning. Thus, some degree of "lossy-ness" allows the network to prioritize visually salient information, further improving accuracy. Furthermore, DCT-based methods exhibited narrower confidence intervals, indicating improved inference precision in the encrypted domain. This improved reliability stems from the -30% reduction in programmable bootstrapping (PBS) (see Table 1), which contribute to error accumulation. Overall, DCT-CryptoNets meets or exceeds the accuracy of RGB-based methods while also enhancing the reliability of private inference.

## 4.3 COMPARATIVE BENCHMARKING OF ACCURACY AND LATENCY

DCT-CryptoNets demonstrates competitive accuracy compared to existing methods on both CIFAR-10 and ImageNet (see Table 4). While CKKS-based methods demonstrate slightly lower latency on smaller models due to their utilization of single instruction multiple data (SIMD) packing, the error accumulation issue of polynomial approximated activation functions (PAAs) hinders CKKS-based methods from scaling to larger networks and high-resolution images like those in ImageNet. It's crucial to note that CKKS's focus on amortized runtime through SIMD packing, a feature unavailable in TFHE, makes direct latency comparisons via normalization less meaningful (see Appendix A.8 for more information on the exact compute resources used for each TFHE-based methods).

Comparing DCT-CryptoNets to SHE (Lou & Jiang, 2019), the only other published method demonstrating scalability to ImageNet, we observe comparable accuracy and latency for smaller models and images. It's worth noting that SHE's fastest model on CIFAR-10 is a custom CNN without skip connections, contributing to its slight latency advantage over DCT-CryptoNets' ResNet-20 model. As network or image size increases, DCT-CryptoNets showcases superior scalability, with progressively greater latency improvements compared to SHE culminating in a 5.3× speedup on ImageNet. Furthermore, even without DCT optimizations, larger DCT-CryptoNets ResNet-18 models outperform SHE on both CIFAR-10 and ImageNet. This improvement can be attributed to SHE's reliance on a leveled TFHE scheme, which necessitates higher multiplicative depths and larger ciphertexts leading to latency amplifications in deeper networks. In contrast, by leveraging programmable bootstrapping, DCT-CryptoNets mitigates these limitations and enables efficient handling of larger models. Isolating the impact of DCT, we compare the latency of DCT-CryptoNets trained on RGB components to DCT-CryptoNets trained on DCT components (see Table 4). Across various model and image sizes, we observe consistent latency improvements ranging from 1.7× to 2.4× when utilizing DCT, independent of training, quantization and cryptographic hyperparameters. Overall, DCT-CryptoNets achieves performance on par with prior work on smaller images and networks, while showcasing superior scalability and efficiency gains when applied to larger, more complex tasks.

Table 4: Performance benchmarking of DCT-CryptoNets. We evaluate performance using the $64\times56^2$ DCT representation for ImageNet and $24\times16^2$ for CIFAR-10, chosen for their superior accuracy. RGB variants of DCT-CryptoNets, trained on $3\times32^2$ or $3\times224^2$ inputs, serve as baselines to isolate the impact of DCT optimizations. DCT-CryptoNets offers comparable performance on smaller models and datasets but excel in scaling to larger ones.

| Method | Dataset | Input Dimension | Model | Scheme | Accuracy | Latency (s) | Normalized Latency (s) (96-threads) |
|---|---|---|---|---|---|---|---|
| Hesamifard et al. (2017) | CIFAR-10 | $3\times32^2$ | Custom CNN | BGV | 91.4% | 11,686 | ~ |
| Chou et al. (2018) | CIFAR-10 | $3\times32^2$ | Custom CNN | FV-RNS | 75.9% | 3,240 | ~ |
| SHE (Lou & Jiang, 2019) | CIFAR-10 | $3\times32^2$ | Custom CNN | TFHE | 92.5% | 2,258 | 470 |
| Lee et al. (2022b) | CIFAR-10 | $3\times32^2$ | ResNet-20 | CKKS | 92.4% | 10,602 | ~ |
| Lee et al. (2022a) | CIFAR-10 | $3\times32^2$ | ResNet-20 | CKKS | 91.3% | 2,271 | ~ |
| Kim & Guyot (2023) | CIFAR-10 | $3\times32^2$ | Plain-20 | CKKS | 92.1% | 368 | ~ |
| Ran et al. (2023) | CIFAR-10 | $3\times32^2$ | ResNet-20 | CKKS | 90.2% | 392 | ~ |
| Rovida & Leporati (2024) | CIFAR-10 | $3\times32^2$ | ResNet-20 | CKKS | 91.7% | 336 | ~ |
| Benamira et al. (2023) | CIFAR-10 | $3\times32^2$ | VGG-9 | TFHE | 74.0% | 570 | 48 |
| Stoian et al. (2023) | CIFAR-10 | $3\times32^2$ | VGG-9 | TFHE | 87.5% | 18,000 | 3,000 |
| **DCT-CryptoNets** | CIFAR-10 | $3\times32^2$ | ResNet-20 | TFHE | 91.6% | 1,339 | 1,339 |
| **DCT-CryptoNets** | CIFAR-10 | $24\times16^2$ | ResNet-20 | TFHE | 90.5% | 565 | 565 |
| SHE (Lou & Jiang, 2019) | CIFAR-10 | $3\times32^2$ | ResNet-18 | TFHE | 94.6% | 12,041 | 2,509 |
| **DCT-CryptoNets** | CIFAR-10 | $3\times32^2$ | ResNet-18 | TFHE | 92.3% | 1,746 | 1,746 |
| **DCT-CryptoNets** | CIFAR-10 | $24\times16^2$ | ResNet-18 | TFHE | 91.2% | 1,004 | 1,004 |
| SHE (Lou & Jiang, 2019) | ImageNet | $3\times224^2$ | ResNet-18 | TFHE | 66.8% | 216,000 | 45,000 |
| **DCT-CryptoNets** | ImageNet | $3\times224^2$ | ResNet-18 | TFHE | 66.1% | 16,115 | 16,115 |
| **DCT-CryptoNets** | ImageNet | $64\times56^2$ | ResNet-18 | TFHE | 66.3% | 8,562 | 8,562 |

## 4.4 LIMITATIONS AND FUTURE DIRECTIONS

Performance is highly dependent on the cryptographic hyperparameters and the degree of bit precision during quantization-aware training (QAT). Careful tuning of precision rounding, programmable boostrapping (PBS) error probability and QAT bit-level is essential to navigate the trade-off between accuracy and latency. Future steps should be made to improve this hyperparameter selection process. Furthermore, the accuracy benefits from DCT are less pronounced on smaller images (e.g., $32\times32$) compared to larger ones. This observation aligns with the inherent nature of small images, which often contain less high-frequency information that DCT filtering typically removes. Consequently, the impact of focusing on low-frequency components is less significant for these smaller inputs. Future work could explore techniques to enhance the accuracy benefits of DCT for smaller images.

## 5 CONCLUSION

In this paper, we presented DCT-CryptoNets, a novel framework that addresses the critical challenges of computational cost and scalability in fully homomorphic encrypted neural networks (FHENNs). By utilizing frequency-domain learning through the Discrete Cosine Transform (DCT), DCT-CryptoNets significantly reduces the computational burden of non-linear activations and homomorphic bootstraps by focusing on low-frequency information, enabling more efficient inference on large-scale datasets like ImageNet.

DCT-CryptoNets exhibits several key advantages: (1) it achieves significant latency improvements, demonstrating up to a $5.3\times$ speedup compared to prior work, with DCT-based optimizations contributing a $1.7\times$ to $2.4\times$ speedup in isolation; (2) focusing on perceptually relevant low-frequency information through DCT maintains or even boosts accuracy compared to RGB-based networks; (3) it enhances the reliability of predictions by reducing error-accumulating homomorphic bootstrap operations, resulting in more precise predictions (e.g., from ±2.5% to ±1.0% on ImageNet); and (4) it demonstrates superior scalability, achieving greater reductions in homomorphic operations compared to RGB-based implementations as network and image size increases.

These advancements showcase the promise of frequency-domain techniques for privacy-preserving deep learning. This not only represents a significant step towards practical deployment of FHE in real-world applications, but also opens new avenues for future research in optimizing private inference for deep learning.

## REPRODUCIBILITY STATEMENT

We are committed to reproducible research and prioritize transparency in our work. To this end regarding explanations in the paper, the Methodology section (Section 3) provides a comprehensive overview of our model topology, training methodology, and encryption process. The Experimental Setup subsection (Section 4.1) offers further details regarding libraries, tools and hardware environments needed to facilitate reproducibility. All training, quantization, and cryptographic hyperparameters are also documented in the Appendix (Appendix A.7). Our code has also been released at https://github.com/ar-roy/dct-cryptonets.

## ACKNOWLEDGEMENTS

This project was supported in part by the Purdue Center for Secure Microelectronics Ecosystem – CSME#210205 and the Center for the Co-Design of Cognitive Systems (CoCoSys), a DARPA-sponsored JUMP 2.0 center. We also acknowledge the insightful discussions with the Zama Concrete-ML team and within the FHE.org community, which enriched our perspective on this research.

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

# A  APPENDIX

## TABLE OF CONTENTS

## A.1  THREAT MODELS AND MITIGATION STRATEGIES

While fully homomorphic encryption (FHE) provides robust security against many traditional attacks, a comprehensive understanding of potential vulnerabilities within the DCT-CryptoNets framework is crucial for ensuring strong privacy guarantees. This analysis must consider the unique security considerations inherent to FHE-based private inference.

- *Server-side security:* The server only receives encrypted DCT coefficients, protecting against a wide range of server-side attacks, including those from an honest-but-curious adversary. However, in the case of a malicious server, additional measures like verifiable computation (VC) may be necessary to ensure correct function evaluation (Viand et al., 2023; Atapoor et al., 2024).

- *Communication-channel security:* Encrypted communication between the client and server safeguards against eavesdropping. However, secure key transfer mechanisms are crucial to prevent unauthorized access to the decryption key.

- *Client-side security:* Data privacy relies heavily on the security of the client device, as this is where the data exists in plaintext. Robust client-side security measures, such as secure key storage and protection against malware, are essential. While FHE itself is secure against various attacks such as chosen/known plaintext attacks and chosen ciphertext attacks (Peikert, 2016), these protections are contingent on the client's ability to safeguard their own secret key.

## A.2 IN THE CONTEXT OF MLaaS

Figure 3: MLaaS system with DCT-CryptoNets.

DCT-CryptoNets leverages the Discrete Cosine Transform (DCT) to operate directly in the frequency domain, aligning with the prevalent use of DCT in image compression standards like JPEG. This enables seamless integration with existing image processing pipelines and facilitates efficient transmission and processing in machine learning as a service (MLaaS) systems and other applications where image data is sent for remote analysis. In this paradigm (see Figure 3), a model is trained locally, encrypted, and then deployed in the cloud. During inference, users transform their images into DCT representations (a process often already performed due to JPEG compression) and encrypt them using their private key. The encrypted image is then sent to the cloud-based model for processing, and the resulting encrypted output is returned to the user. To obtain the final prediction, the user decrypts the output of the penultimate layer (prior to the fully-connected layer) and processes the resulting plaintext tensor through a local classifier.

In many FHE-based private inference applications, both a *secret* key and *evaluation* key are generated. The *evaluation* key, analogous to a public key specifying the operations of the FHE circuit, allows for the execution of computations on the encrypted model. This approach ensures that only the user possessing the corresponding secret key can decrypt the resulting inference output:

- *Deployment:* A trained fully homomorphic encrypted neural network (FHENN) is deployed to the cloud server.
- *User request:* User requests the FHENN cryptographic parameters. Once received they generated both *secret* and *evaluation* keys locally. Each client would therefore have their own *secret* key.
- *Key Exchange:* User sends the *evaluation* key along with their encrypted inputs to the server.
- *Inference:* Server runs private inference with the users' *evaluation* key and encrypted inputs. Then sends encrypted results back to the user.
- *User decryption:* User then decrypts the results with their *secret* key.

## A.3 ReLU Pruning Techniques

Table 5: Impact of different ReLU pruning techniques ($3\times224^2$ RGB $\rightarrow$ $64\times56^2$ DCT).

| $\Delta$: RGB $\rightarrow$ DCT | | DCT-Net (Xu et al., 2020) | + First Layer Pruned (Ghodsi et al., 2020) | + Skip Connections Pruned (Ghodsi et al., 2020) | + Both Pruned |
|---|---|---|---|---|---|
| Ops. | #ReLUs | -26.1% | **-34.8%** | -58.7% | -67.4% |
| | #HOPs | -21.7% | **-33.1%** | -24.3% | -25.7% |
| Acc. | CIFAR-10 | +0.0% | **-0.4%** | -2.67% | -2.54% |
| | ImageNet | +2.2% | **+2.3%** | $\sim$ | $\sim$ |

DCT-Net (Xu et al., 2020) inherently reduces ReLU operations due to spatial dimension reduction. DCT-CryptoNets further optimizes latency by pruning the first layer's ReLU's, with minimal impact on accuracy. However, skip connection ReLU pruning is not recommended as it incurs significant accuracy degradation and increases homomorphic operations even though the number of ReLU operations decreases. This is because removing ReLU after the addition of two quantized tensors necessitates the insertion of a quantization identity function in TFHE-based networks.

## A.4 DCT-CryptoNets for Small(er) Images and Networks

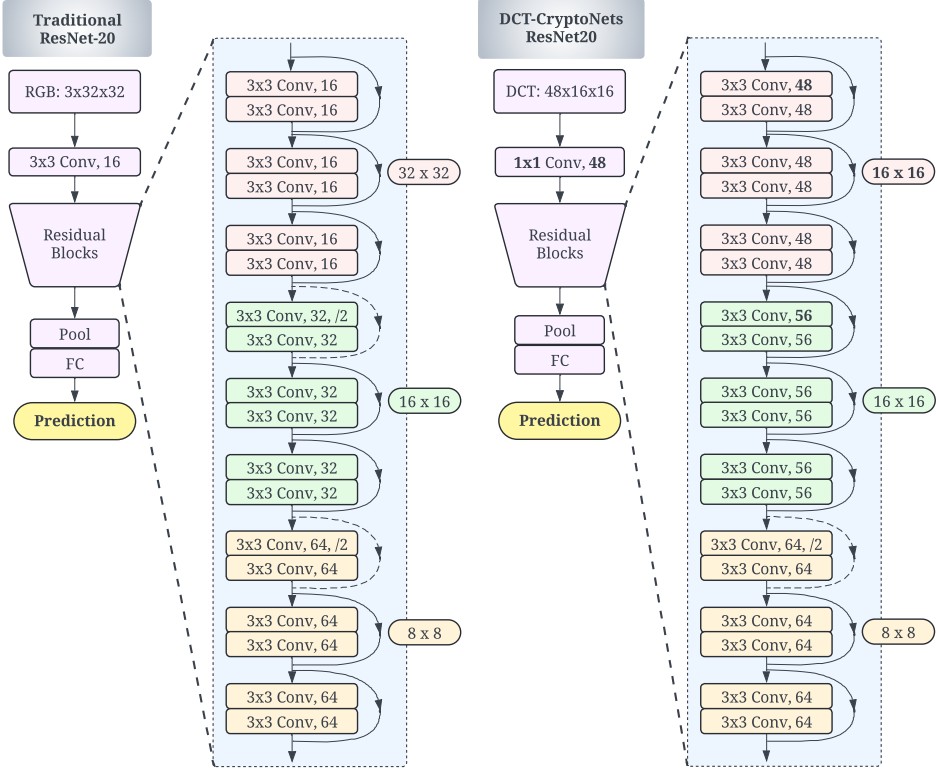

Figure 4: Traditional vs. DCT-CryptoNets ResNet-20 Architecture. Changes between the two architectures are bolded.

When applying DCT optimizations to smaller images and networks (e.g., ResNet-20 on $32\times32$ images), adaptations are necessary to address the excessive spatial dimension reduction caused by the standard $8\times8$ DCT filter. We mitigate this by reducing the DCT filter size to $4\times4$, resulting in

48 frequency channels (16 per YCrCb component) instead of the typical 192. To accommodate this change, we introduce a modified ResNet-20 architecture with two key adjustments (see Figure 4):

- *Downsampling Reduction:* One downsampling step is removed to preserve spatial resolution throughout the network.
- *Channel Expansion:* The initial channel dimensions are increased to match the 48-channel DCT output, preventing information loss during the first convolutional layer transition. The modified channel dimensions are now [48, 56, 64].

Although this channel expansion doubles the number of parameters, it enables effective DCT utilization on smaller images. As demonstrated in Table 2, DCT-CryptoNets ResNet-20 achieves a 4% accuracy improvement over the traditional ResNet-20 while maintaining latency.

## A.5   QUANTIZATION ABLATION ON IMAGENET

Table 6: Quantization impact for ResNet-18 on ImageNet

|  | Quantization | Accuracy | Latency (s) |
|---|---|---|---|
| RGB ($3{\times}224^2$) | 4-bit | 60.6% | 12,874 |
|  | 5-bit | 66.1% | 16,115 |
|  | 8-bit | 68.4% | 75,674 |
| DCT ($64{\times}56^2$) | 4-bit | 62.9% | 6,961 |
|  | 5-bit | 66.3% | 8,562 |
|  | 8-bit | 68.5% | 41,998 |

To examine the impact of quantization on performance, we varied the quantization levels on ImageNet, as detailed in Table 6. Adopting 5-bit quantization, consistent with the approach in SHE (Lou & Jiang, 2019), resulted in a notable accuracy improvement compared to 4-bit quantization, with a manageable increase in latency of approximately 25%. Further increasing the quantization level to 8-bit yielded marginal accuracy gains, while incurring a substantial increase in latency. These results suggest that 5-bit quantization strikes an effective balance between accuracy and efficiency for large-scale datasets like ImageNet.

## A.6   SENSITIVITY OF CRYPTOGRAPHIC HYPERPARAMETERS IN FHENN

Table 7: Comparing the change of latency, top-1 accuracy and ciphertext memory requirements of a $3{\times}224^2$ model on 1000 samples from ImageNet when modulating cryptographic hyperparameters (precision rounding and PBS error probability).

| PBS Err. | Precision Rounding = 6 | | | Precision Rounding = 7 | | | Precision Rounding = 8 | | |
|---|---|---|---|---|---|---|---|---|---|
|  | Δ Latency | Δ Acc. | Δ Memory | Δ Latency | Δ Acc. | Δ Memory | Δ Latency | Δ Acc. | Δ Memory |
| 0.05 | -25.6% | -36.8% | -24.4% | -10.7% | -32.2% | -17.9% | +34.0% | -35.7% | +19.1% |
| 0.01 | -12.1% | -21.3% | -6.2% | ★ | ★ | ★ | +81.7% | +0.4% | +56.8% |
| 0.005 | -34.2% | -9.4% | -8.0% | +62.3% | +0.0% | +5.5% | +203.4% | +1.2% | +93.3% |

Encryption of the neural network requires careful selection of cryptographic hyperparameters to balance accuracy and latency. Starting with our pre-selected values for ImageNet (precision rounding of 7 and PBS error probability of 0.01), as detailed in Table 7, we systematically explore the effects of varying these parameters. Reducing either parameter leads to a significant drop in top-1 accuracy. Increasing these parameters generally yields a noticeable increase in latency, with only marginal accuracy gains. Notably, raising the bit-precision rounding to 8 and halving the PBS probability increases top-1 accuracy on ImageNet subsamples by up to 1.2% but incurs a threefold increase in latency.

## A.7 TRAINING, QUANTIZATION AND CRYPTOGRAPHIC HYPERPARAMETERS

Table 8: Training, quantization and cryptographic hyperparameters.

| Section | Hyperparameter | CIFAR-10 mini-ImageNet Imagenette | ImageNet |
|---|---|---|---|
| Training | Epochs | 60 | 90 |
| | Batch Size | 32 | 256 |
| | Optimizer | Adam | Adam |
| | Learning Rate | 1e-3 | 1e-3 |
| | Weight Decay | 1e-5 | 1e-5 |
| | Gradient Clipping | 0.1 | 0.1 |
| | Dropout | 0.2 | 0.2 |
| | Scheduler | [20, 45] | [30, 60] |
| | Decay Factor | 0.1 | 0.1 |
| Quantization | Weight Bit-Width | 4 | 4 |
| | Weight Quantization Protocol | Int8WeightPerTensorFloat[‡] | Int8WeightPerTensorFloat[‡] |
| | Activation Bit-Width | 4 | 4 |
| | Activation Quantization Protocol | Int8ActPerTensorFloat[‡] | Int8ActPerTensorFloat[‡] |
| Cryptographic | Number of Bits | 5 | 5 |
| | PBS Error Probability | 0.01 | 0.01 |
| | Bit-removal Rounding Threshold | 6 | 7 |

## A.8 TFHE-BASED FHENN COMPUTE RESOURCES

Table 9: Comparison of TFHE-based methods based on available compute resources.

| Method | Dataset | Input Dimension | Model | Accuracy | Reported Latency (s) | CPU Threads | Normalized Latency (s) (96-threads) |
|---|---|---|---|---|---|---|---|
| SHE (Lou & Jiang, 2019) | CIFAR-10 | $3\times32^2$ | Custom CNN | 92.5% | 2,258 | 20 | 470 |
| Benamira et al. (2023) | CIFAR-10 | $3\times32^2$ | VGG-9 | 74.01% | 570 | 8 | 48 |
| Stoian et al. (2023) | CIFAR-10 | $3\times32^2$ | VGG-9 | 87.5% | 18,000 | 16 | 3,000 |
| **DCT-CryptoNets** | CIFAR-10 | $3\times32^2$ | ResNet-20 | 91.6% | 1,339 | 96 | 1,339 |
| **DCT-CryptoNets** | CIFAR-10 | $24\times16^2$ | ResNet-20 | 90.5% | 565 | 96 | 565 |
| SHE (Lou & Jiang, 2019) | CIFAR-10 | $3\times32^2$ | ResNet-18 | 94.62% | 12,041 | 20 | 2,509 |
| **DCT-CryptoNets** | CIFAR-10 | $3\times32^2$ | ResNet-18 | 92.3% | 1,746 | 96 | 1,746 |
| **DCT-CryptoNets** | CIFAR-10 | $24\times16^2$ | ResNet-18 | 90.9% | 1,004 | 96 | 1,004 |
| **DCT-CryptoNets** | CIFAR-10 | $3\times224^2$ | ResNet-18 | 92.4% | 11,097 | 96 | 11,097 |
| **DCT-CryptoNets** | CIFAR-10 | $64\times56^2$ | ResNet-18 | 92.4% | 6,313 | 96 | 6,313 |
| SHE (Lou & Jiang, 2019) | ImageNet | $3\times224^2$ | ResNet-18 | 69.4% | 216,000 | 20 | 45,000 |
| **DCT-CryptoNets (4-bit)** | ImageNet | $3\times224^2$ | ResNet-18 | 60.6% | 12,874 | 96 | 12,874 |
| **DCT-CryptoNets (5-bit)** | ImageNet | $3\times224^2$ | ResNet-18 | 66.1% | 16,115 | 96 | 16,115 |
| **DCT-CryptoNets (4-bit)** | ImageNet | $64\times56^2$ | ResNet-18 | 62.9% | 6,961 | 96 | 6,961 |
| **DCT-CryptoNets (5-bit)** | ImageNet | $64\times56^2$ | ResNet-18 | 66.3% | 8,562 | 96 | 8,562 |

Table 9 presents latency values normalized to the 96-thread computational capacity of our DCT-CryptoNets implementation. This normalization is justified for TFHE-based schemes, which prioritize individual inference latency, as the Concrete-ML library fully utilizes available CPU resources. Our experiments on a 24-thread Intel Core i9-12900KF confirmed a linear relationship between core count and latency, with a 4x increase observed compared to the 96-thread dual AMD Ryzen Threadripper PRO 5965WX. In contrast, CKKS-based methods often leverage single instruction multiple data (SIMD) packing for amortized runtime optimization, a feature not available in TFHE. Consequently, direct latency normalization may not accurately reflect the performance characteristics of CKKS-based approaches due to their inherent reliance on batch processing.

[‡]Recommended quantization protocols by Concrete-ML for bit-width > 3.

