# OpenReview forum: "DCT-CryptoNets: Scaling Private Inference in the Frequency Domain"
_ICLR.cc/2025/Conference — ICLR 2025 Poster_

### Official Review · Reviewer_EoRi · 2024-11-03

**Soundness:** 3
**Presentation:** 3
**Contribution:** 3
**Rating:** 6
**Confidence:** 3

**Summary:**

The paper presents DCT-CryptoNets, a novel framework that enhances privacy-preserving deep learning by utilizing the Discrete Cosine Transform (DCT) to operate in the frequency domain, significantly reducing latency and computational costs associated with fully homomorphic encrypted neural networks (FHENNs). By focusing on low-frequency information, DCT-CryptoNets improve accuracy and reliability in encrypted predictions while demonstrating superior scalability with increasing image resolution. The work addresses key challenges in existing FHE-based neural networks and emphasizes the importance of optimizing cryptographic and quantization parameters for practical applications in secure image processing.

**Strengths:**

1. **Frequency-Domain Optimization**: DCT-CryptoNets leverage the Discrete Cosine Transform (DCT) to focus on low-frequency components of images, which enhances the model's ability to capture perceptually salient information while reducing computational complexity and improving accuracy compared to traditional RGB-based networks.

2. **Reduced Latency and Improved Scalability**: The proposed method achieves significant latency reductions (up to 5.3×) during inference, especially on large datasets like ImageNet, while demonstrating superior scalability as image resolution increases. This makes DCT-CryptoNets more efficient for real-world applications of privacy-preserving deep learning.

3. **Enhanced Reliability through Reduced Error Accumulation**: By minimizing the need for homomorphic bootstrap operations, DCT-CryptoNets reduce the accumulation of approximation errors, leading to more reliable predictions and improved encrypted accuracy. This addresses challenges faced by earlier fully homomorphic encryption (FHE) schemes.

**Weaknesses:**

I have a question about the threat model setting here, why the model is trained locally?

If the client could train the model by themselves, why does they do inference locally.

Or if they want to deploy that encrypted model (key from model trainer) on the cloud for other clients as service. How does the key management should be solved?

**Questions:**

see weakness

---

> ### Author Response · Authors · 2024-11-19
>
> Dear EoRi,
>
> We are grateful for your insightful feedback and careful consideration of our work. We hope that the following answers all of the questions that you had regarding this work:
>
> ---
>
> ### Response to Weaknesses/Questions
>
> > Q1/Q2: … why is the model trained locally? … If the client could train the model by themselves, why do they do inference locally?
>
> -   DCT-CryptoNets employs a practical approach to private inference by performing model training locally on plaintext data. This strategy is motivated by the significant computational overhead associated with homomorphic training, which remains a challenging research area. Given that FHE-based inference on a single image can take several minutes, training a large model directly in the encrypted domain would be prohibitively time-consuming, potentially requiring weeks or even months.
>
> -   However, this local training approach aligns well with the paradigm of inference model as a service (MLaaS), where a company or organization leverages its own secure infrastructure to train a model. The trained model is then encrypted and deployed in a potentially less trusted environment, such as the cloud, to provide private inference services to users. This separation of training and inference not only addresses the current limitations of homomorphic training but also enhances the practicality of deploying privacy-preserving models in real-world settings, ensuring both model confidentiality and user data privacy.
>
> > Q3: If they want to deploy the encrypted model on the cloud for other clients as a service. How is the key management solved?
>
> - We appreciate you raising this important consideration. In many private inference applications utilizing FHE, both a *secret* key and *evaluation* key are generated. The *evaluation* key can be seen as a specific type of public key which can be used to run a trained fully homomorphic encrypted neural network (FHENN). This approach ensures that only the user possessing their own *secret* key can decrypt their results:
>
> - Steps:
>
> 	1. A trained FHENN is deployed to the cloud server.
>
> 	2. User requests the FHENN cryptographic parameters. Once received they generated both the *secret* and *evaluation* keys locally. Each client would therefore have their own secret key.
>
> 	3. User sends the *evaluation* key along with their encrypted inputs to the server.
>
> 	4. Server runs private inference with the users’ *evaluation* key and encrypted inputs. Then sends encrypted results back to the user.
>
> 	5. User then decrypts the results with their *secret* key.
>
> - We have included more detail in a pre-existing appendix section (Section A.2) which elaborates on these questions.

---

> ### Author Response · Authors · 2024-11-24
> **Nearing the end of the discussion period**
>
> Dear EoRi,
>
> We hope that you're doing well. Thank you again for your careful consideration of this work. As the discussion period is coming to an end, we are curious to know if our clarifications and additions to the manuscript have sufficiently addressed your questions. We welcome any further questions or discussions you may have!
>
> Best,
> Authors

---

> > ### Author Response · Authors · 2024-12-02
> >
> > Dear EoRi,
> >
> > Thank you for your time and detailed feedback on this work. As the discussion period ends today (updated from Nov. 26th to Dec. 2nd), we would like to double check if we have adequately addressed all your concerns. We have incorporated your feedback regarding key-management into the revised manuscript and hope that our responses and updates have successfully addressed all the points you raised.
> >
> > Thank you again for your valuable time and constructive feedback that has helped enhance the quality of our work!
> >
> > Best,\
> > Authors

---

### Official Review · Reviewer_SG3z · 2024-11-03

**Soundness:** 4
**Presentation:** 4
**Contribution:** 4
**Rating:** 8
**Confidence:** 4

**Summary:**

This paper presents a very promising strategy for achieving a practical and computationally efficient FHE-based (CKKS) private inference of deep learning models applied to high resolution images.     In a novel approach, the technique presented relies as a starting point on a Discrete Cosine Transform representation of the image.  It also employs a quantization aware training (QAT) framework.  Thorough comparative benchmarking of accuracy and latency for RGB vs DCT techniques a various image resolutions and at various levels of retained low-frequency components (of the DCT) are presented.

The paper establishes both a detailed methodology and new "high-water mark" performance capability for ML inference generation for high resolution images.  Though performance is less pronounced on smaller images (32x32) it is the capability on larger images that is most important for future practical applications.

**Strengths:**

Novel DCT based insight for unstructured data yielding excellent computational advantages for high resolution images.

Thorough benchmarking and attention to reproducible science.

Excellent comparative analysis of this CKKS based technique relative to competing TFHE approaches.

Excellent and comprehensive list of references that chronicle the current state of the art and prior advances.

**Weaknesses:**

I find no obvious weaknesses.

I would emphasize to the reader perhaps new to the field ( in Section 3.2 on page 6)  that model training is done in the plaintext domain. (even though this is most evident in Figure 3 presented on Page 14.

**Questions:**

Are there any special scaling or normalization techniques that need to be considered when considering the YCrCb color space components of an image?

Can GPUs be used to any advantage in this approach?

Are there any privacy preservation guarantees or "leakage" guarantees that could be developed around this approach.  (I realize this may be challenging mathematically.)

---

> ### Author Response · Authors · 2024-11-19
>
> Dear SG3z,
>
> Thank you for your thoughtful review. We particularly appreciate your understanding of the key strengths of our approach. We would like to point out a potential typo and clarify that our method employs TFHE. Besides this, we hope that the following answers the questions that you had regarding our work:
>
> ---
>
> ### Response to Questions
>
> > Q1: Are there any special scaling or normalization techniques that need to be considered when considering the YCrCb color space components of an image?
>
> - Indeed, normalization is necessary when working with DCT coefficients, just as it is for other image representations. We follow the standard practice of normalizing the tensors using the mean and variance calculated from the full training dataset. However, since pre-computed mean and variance values for DCT coefficients on benchmark image datasets are not readily available, these statistics need to be calculated beforehand.
>
> - This normalization was not explicitly mentioned in the “Methodology” text but can be seen in the last step of the “DCT Encoding” section in Figure 2. This is also most evident in the provided anonymous code repository.
>
> > Q2: Can GPUs be used to any advantage in this approach?
>
> - GPUs can in fact be leveraged to accelerate DCT-CryptoNets. Recent advancements in FHE libraries, such as the introduction of GPU support in Concrete-ML v1.7.0 (released Sept. 2024), are already demonstrating the potential for latency improvements. While initial benchmarks show a 1-2x speedup, we anticipate even greater gains as these libraries mature and incorporate further optimizations. This aligns with existing research showcasing substantial speedups (up to 20x or more) for TFHE on GPUs [1, 2].
>
> - Furthermore, initiatives like the DARPA DPrive program, focused on developing dedicated homomorphic hardware accelerators, hold the promise of even more dramatic latency improvements (up to 1,000x). We believe that hardware acceleration, both through GPUs and specialized hardware, will be crucial for the widespread adoption of privacy-enhancing technologies like fully homomorphic encrypted neural networks (FHENNs).
>
> > Q3: Are there any privacy preservation guarantees or "leakage" guarantees that could be developed around this approach. (I realize this may be challenging mathematically.)
>
> - Thank you for bringing this important point up. While FHE offers robust protection against many traditional attacks, understanding the potential vulnerabilities and attack surfaces in the context of DCT-CryptoNets is crucial for ensuring the system's overall security and privacy guarantees. Many of these considerations are specifically focusing on the security considerations unique to FHE-based private inference.
>
> - *Server-Side Attacks:* The server only receives encrypted DCT coefficients, protecting against a wide range of server-side attacks, including those from an honest-but-curious adversary. However, in the case of a malicious server, additional measures like verifiable computation (VC) may be necessary to ensure correct function evaluation [3, 4]. Integrating VC with FHE is an active area of research.
>
> - *Communication Channel Security:* Encrypted communication between the client and server safeguards against eavesdropping. However, secure key transfer mechanisms are crucial to prevent unauthorized access to the decryption key.
>
> - *Client-Side Vulnerabilities:* Data privacy relies heavily on the security of the client device, as this is where the data exists in plaintext. Robust client-side security measures, such as secure key storage and protection against malware, are essential. While FHE itself is secure against various attacks such as chosen/known plaintext attacks and chosen ciphertext attacks [5], these protections are contingent on the client's ability to safeguard their own secret key.
>
> - We have added a new appendix section (Section A.1) which details all of the above threat models and mitigation strategies.
>
> ---
>
> ### References
>
> [1] Morshed et al., CPU and GPU accelerated fully homomorphic encryption, IEEE HOST, 2020.
>
> [2] Wang et al., HE-Booster: An efficient polynomial arithmetic acceleration on GPUs for fully homomorphic encryption. IEEE Transactions on Parallel and Distributed Systems, 2023.
>
> [3] Viand et al., Verifiable fully homomorphic encryption, Arxiv, 2023.
>
> [4] Atapoor et al., Verifiable FHE via lattice-based SNARKs, IACR Communications in Cryptology, 2024.
>
> [5] Chris Peikert. A decade of lattice cryptography, Found. Trends Theor. Comput. Sci., 2016

---

> > ### Comment · Reviewer_SG3z · 2024-11-24
> > **Response to official comment by authors**
> >
> > I am fully satisfied with the response provided by the authors and their proposed clarifications and revision to the manuscript.

---

> > > ### Author Response · Authors · 2024-11-26
> > >
> > > Dear SG3z,
> > >
> > > Thank you very much for taking the time to confirm our responses. We sincerely appreciate your recognition of our work and your valuable suggestions, which have contributed to strengthening our paper.
> > >
> > > Best,\
> > > Authors

---

> ### Author Response · Authors · 2024-11-24
> **Nearing the end of the discussion period**
>
> Dear SG3z,
>
> We hope that you're doing well. Thank you again for thoughtful review of our submission. As the discussion period is coming to an end, we are curious to know if our responses to your questions were sufficiently addressed. We welcome any further questions or discussions you may have!
>
> Best,
> Authors

---

### Official Review · Reviewer_qW2q · 2024-11-08

**Soundness:** 3
**Presentation:** 3
**Contribution:** 2
**Rating:** 5
**Confidence:** 4

**Summary:**

This paper introduces DCT-CryptoNets, which performs neural network (CNN) inference entirely using TFHE-based homomorphic encryption in the frequency domain. Unlike hybrid protocols that limit homomorphic encryption to just linear operations, and require MPC for nonlinear operations, this approach leverages TFHE to perform both linear and nonlinear operations under homomorphic encryption. This enables private inference outsourcing to the server without requiring client interaction for intermediate computations. The authors have shown significant speedup over prior TFHE-based implementation (SHE, NeurIPS'19) on CIFAR-10 and ImageNet datasets.

**Strengths:**

$\bullet$ Performing TFHE-based inference in the frequency domain allows the usage of lower-resolution inputs without compromising accuracy. This approach significantly decreases both FLOPs and nonlinear operations (ReLU), while also requiring fewer bootstrapping operations. This results in substantial speedup benefits. More importantly, this makes it feasible to perform inference on larger input images, enhancing the practical applicability (such as semantic segmentation) of homomorphic encryption in neural network inference.


$\bullet$ The authors report ciphertext accuracy, specifically on the ImageNet-1K scale, distinguishing this work from most private inference papers, which typically report plaintext accuracy under the assumption that there is no accuracy loss when operations are conducted in field arithmetic.

$\bullet$ The experimental results are extensive and include a detailed sensitivity analysis of the cryptographic hyper-parameters.

**Weaknesses:**

$\bullet$ The lack of sufficient algorithmic contributions and research insights makes it less suitable for the ML conference. Operating in the frequency domain for private inference benefits is not a novel concept (see [1,2]). Also, the usage of quantization-aware training for lower-frequency components is simply an engineering tweak.

Thus, a more fitting venue for this work might be a cryptography-focused conference.

$\bullet$ Moreover, the practicality of HE-only private inference remains questionable, especially when compared to hybrid protocol-based approaches. For example, Cheetah [3] achieves ImageNet-1K inference on ResNet-50 in 80.3 seconds in a LAN setting and 134.7 seconds in a WAN setting. Thus, the primary motivation for pursuing HE-only inference appears to be the benefit of non-interactive private inference.




1. Song et al., ENSEI: Efficient secure inference via frequency-domain homomorphic convolution for privacy-preserving visual recognition, CVPR 2020.

2. Li et al. Falcon: A fourier transform based approach for fast and secure convolutional neural network predictions, CVPR 2020.

3. Huang et al., Cheetah: Lean and fast secure Two-Party deep neural network inference, USENIX Security 2022.

**Questions:**

See the weakness.

---

> ### Author Response · Authors · 2024-11-19
>
> Dear qW2q,
>
> We sincerely appreciate your feedback and the time you invested in reviewing our submission. Your constructive feedback provides both encouragement and practical guidance for strengthening this work. We are eager to respond to your questions and comments:
>
> ---
>
> ### Response to Weaknesses/Questions
>
> > W1.1: The lack of sufficient algorithmic contributions and research insights makes it less suitable for the ML conference… Operating in the frequency domain for private inference benefits is not a novel concept (see [1,2]) … Thus, a more fitting venue for this work might be a cryptography-focused conference.
>
> - We appreciate your feedback regarding the contributions and insights of our work. While individual components have definitely been explored, our work aims to bridge the gap towards practical privacy-preserving learning on large-scale networks and images. While fundamentally a cryptographic solution, our core focus and impact lie in advancing privacy in ML, making this contribution well-suited for this venue.
>
> - We acknowledge prior work on frequency-domain techniques for private inference, such as those utilizing the homomorphic discrete Fourier transform (HDFT) for the BFV [2] and CKKS schemes [4]. These methods, while introducing specialized convolutional blocks for frequency-domain processing within the homomorphic circuit, often exhibit limitations in accuracy and scalability. Such as Falcon [2] achieving 76.5% on CIFAR-10, ENSEI [1] achieving 82.0% on CIFAR-10 and Kim et al. [4] scaling to a Plain-18 but only encrypting the last 8 layers when operating on ImageNet.
>
> - In contrast, DCT-CryptoNets learns features from the DCT representation of images directly, enabling the network to learn from the rate of change in intensities. It does so without specialized convolution operators, enabling seamless integration with existing CNN architectures, achieves much higher accuracy than existing HDFT methods, and shows scalability to ImageNet, a feat unmatched by other frequency-domain private inference methods.
>
> - To address concerns about novelty and relevance to the ML domain, we've clarified the distinctions between DCT-CryptoNets and existing frequency-domain private inference techniques in Sections 2.2 and 2.5 (highlighted in red). We believe these revisions better demonstrate the contributions and applicability of our work within an ML context.
>
> > W1.2: … Also, the usage of quantization-aware training for lower-frequency components is simply an engineering tweak. …
>
> - We appreciate your perspective. However, we believe that quantization-aware training (QAT) is not merely an engineering tweak but a crucial component for optimizing TFHE-based FHENNs. TFHE computations fundamentally rely on Boolean and integer logic gates, making QAT essential for aligning network parameters with this representation to achieve high accuracy.
>
> > W2: The practicality of HE-only private inference remains questionable, especially when compared to hybrid protocol-based approaches … Thus, the primary motivation for pursuing HE-only inference appears to be the benefit of non-interactive private inference.
>
> - We agree that currently there are performance gaps between fully homomorphic encryption (FHE) and hybrid private inference methods. While hybrid approaches may offer better efficiency today, non-interactive FHE offers superior security, particularly in zero-trust environments or under threats like malicious servers. This enhanced security stems from FHE's ability to maintain constant encryption throughout the computation, unlike interactive hybrid methods that involve potentially risky decryption of intermediate values.
>
> - The current state of FHENNs is indeed largely confined to smaller images and neural networks. Our work is a stepping stone that aims to bridge the efficiency gap in FHE methods, facilitating their wider adoption. We anticipate that the combined progress in ML algorithms and systems, akin to this work, as well as improvements in homomorphic hardware acceleration will ultimately establish FHE as a compelling and practical solution for privacy-preserving ML inference applications.
>
> ---
>
> ### References
>
> [1] Song et al., ENSEI: Efficient secure inference via frequency-domain homomorphic convolution for privacy-preserving visual recognition, CVPR 2020.
>
> [2] Li et al., Falcon: A fourier transform based approach for fast and secure convolutional neural network predictions, CVPR 2020.
>
> [3] Huang et al., Cheetah: Lean and fast secure Two-Party deep neural network inference, USENIX Security 2022.
>
> [4] Kim et al., Optimized privacy-preserving CNN inference with fully homomorphic encryption. IEEE Transactions on Information Forensics and Security, 2023.

---

> ### Author Response · Authors · 2024-11-24
> **Nearing the end of the discussion period**
>
> Dear qW2q,
>
> We hope that you're doing well. Thank you again for the detailed and constructive feedback during the review process. As the discussion period is coming to an end, we are curious to know if our clarifications and additions have helped address some of the points you brought up. We welcome any further questions or discussions you may have!
>
> Best,
> Authors

---

> > ### Comment · Reviewer_qW2q · 2024-11-25
> > **Response on the Author's Rebuttal**
> >
> > Thanks for the rebuttal.
> >
> > In my view, the key contribution of this paper is scaling TFHE to ImageNet (which is not done by prior work on TFHE), by using DCT techniques and learning features in the frequency domain (which enables the users to reduce the resolution of the input image, hence reduce the number of linear and non-linear operations).
> >
> > Given these contributions, I will maintain my score, as I still believe that there is a lack of substantial (algorithmic) contributions. In particular, I have not learned anything new from this paper, and I believe that it does not enhance the knowledge sphere of private inference.  Nonetheless, the authors will get the credits for scaling TFHE to ImageNet.
> >
> > Now it's up to the area chair to evaluate the suitability of this paper to ICLR.

---

> > > ### Author Response · Authors · 2024-11-26
> > >
> > > Dear qW2q,
> > >
> > > Thank you for your further consideration and feedback. We appreciate your acknowledgment of our achievement in scaling TFHE-based fully homomorphic encrypted neural networks (FHENN's) to ImageNet.
> > >
> > > **Contributions to the ICLR community:** We believe this work contributes relevant and impactful knowledge to the ICLR community by demonstrating a practical and efficient solution for FHENN's that provides significant computational advantages for high-resolution images, with promising ramifications for images with even higher resolution than ImageNet. Our approach has positive implications for private inference in machine learning, particularly for real-world applications. The comprehensive empirical evidence presented supports the value of our contribution to the ICLR community, advancing the development of efficient and scalable FHE-based deep learning via the systematic learning of low-frequency DCT components.
> > >
> > > We really appreciate the time and effort put into reviewing our work as well as the on-going rebuttal engagement.\
> > > Best,\
> > > Authors

---

### Author Response · Authors · 2024-11-19
**Revision**

We sincerely thank all of the reviewers for their insightful comments and the opportunity to strengthen our work. In the following, we present the changes made to this revision which are marked in red.

-   We have included more clarity on the differences between DCT-CryptoNets and other frequency-domain private inference techniques (mainly leveraging homomorphic discrete Fourier transform - HDFT) in Sections 2.2 and 2.5. This is in response to qW2q comment on other frequency-domain private inference techniques.

-   We have added a new appendix section (Section A.1) which discusses different threat models for which DCT-CryptoNets protects against from a server, client and communication standpoint. We also briefly go into mitigation strategies for certain attacks. This is in response to SG3z’s question on privacy preservation guarantees that could be built upon DCT-CryptoNets.

-   We have included more detail in a pre-existing appendix section (Section A.2) which elaborates on how key management should be handled in FHE-based private inference applications. This is in response to EoRi’s question on how key’s should be managed in DCT-CryptoNets.

We hope that these new changes as well as the individual review responses enhance the clarity, rigor, and impact of this work. We are more than happy to answer any follow-up questions.

---

### Meta-Review · Area_Chair_wQmY · 2024-12-24

**Metareview:**

The paper proposes a new technique DCT-Cryptonets in order to be able to perform cryptographically private (using Fully Homomorphic Encryption) inference on neural networks, particularly for convolutional neural networks. The main contribution is the method to perform operations in frequency space which has benefits both computationally and to a lesser extent statistically due to focus on low-dimensional Fourier transforms. The authors back up their new method by an extensive empirical evaluation and in particular report results on a scale that have not been reported before.

**Additional Comments On Reviewer Discussion:**

Most of the reviewers engaged well with the authors. The main objection of reviewer qW2q was that there wasn't an algorithmic contribution and that the paper may be better suited for a crypto conference. There wasn't much discussion in response to my question; however, I disagree with reviewer qW2q. The paper is certainly not suitable for a crypto conference as there is no novelty on the crypto front. I think there are contributions on the ML side that make it suitable for an ML conference or a Systems conference. Similar work has appeared at ICML/ICLR/NeurIPS and I recommend acceptance.

---

### Decision · Program_Chairs · 2025-01-22

Accept (Poster)